# Ciprofloxacin and Levofloxacin as Potential Drugs in Genitourinary Cancer Treatment—The Effect of Dose–Response on 2D and 3D Cell Cultures

**DOI:** 10.3390/ijms222111970

**Published:** 2021-11-04

**Authors:** Tomasz Kloskowski, Kamil Szeliski, Zuzanna Fekner, Marta Rasmus, Paweł Dąbrowski, Aleksandra Wolska, Natalia Siedlecka, Jan Adamowicz, Tomasz Drewa, Marta Pokrywczyńska

**Affiliations:** Chair of Urology and Andrology, Department of Regenerative Medicine, Cell and Tissue Bank, Collegium Medicum, Nicolaus Copernicus University, 85-094 Bydgoszcz, Poland; kamil.szeliski@cm.umk.pl (K.S.); zuzanna.fekner@cm.umk.pl (Z.F.); marta.rasmus@gmail.com (M.R.); paweldobrowski@gmail.com (P.D.); aleksandra.wolska96@gmail.com (A.W.); siedleckanatalia92@gmail.com (N.S.); adamowicz.jz@gmail.com (J.A.); zmr@cm.umk.pl (T.D.); marta.pokrywczynska@interia.pl (M.P.)

**Keywords:** 3D cell culture, ciprofloxacin, levofloxacin, drug repositioning, bladder cancer, prostate cancer, topoisomerase

## Abstract

Introduction: Introducing new drugs for clinical application is a very difficult, long, drawn-out, and costly process, which is why drug repositioning is increasingly gaining in importance. The aim of this study was to analyze the cytotoxic properties of ciprofloxacin and levofloxacin on bladder and prostate cell lines in vitro. Methods: Bladder and prostate cancer cell lines together with their non-malignant counterparts were used in this study. In order to evaluate the cytotoxic effect of both drugs on tested cell lines, MTT assay, real-time cell growth analysis, apoptosis detection, cell cycle changes, molecular analysis, and 3D cultures were examined. Results: Both fluoroquinolones exhibited a toxic effect on all of the tested cell lines. In the case of non-malignant cell lines, the cytotoxic effect was weaker, which was especially pronounced in the bladder cell line. A comparison of both fluoroquinolones showed the advantage of ciprofloxacin (lower doses of drug caused a stronger cytotoxic effect). Both fluoroquinolones led to an increase in late apoptotic cells and an inhibition of cell cycle mainly in the S phase. Molecular analysis showed changes in *BAX*, *BCL2*, *TP53*, and *CDKN1* expression in tested cell lines following incubation with ciprofloxacin and levofloxacin. The downregulation of topoisomerase II genes (*TOP2A* and *TOP2B*) was noticed. Three-dimensional (3D) cell culture analysis confirmed the higher cytotoxic effect of tested fluoroquinolone against cancer cell lines. Conclusions: Our results suggest that both ciprofloxacin and levofloxacin may have great potential, especially in the supportive therapy of bladder cancer treatment. Taking into account the low costs of such therapy, fluoroquinolones seem to be ideal candidates for repositioning into bladder cancer therapeutics.

## 1. Introduction

Quinolones are a group of antibiotics with anticancer properties. They are divided into four groups; a fluoride atom was added to drug molecules in second quinolone group, leading to the formation of fluoroquinolones. Their mechanism of action is based on an inhibition of bacterial gyrase [1,2]. They are very effective against Gram-negative bacteria and most of their representatives accumulates in urine and organs such as the prostate, reaching concentrations higher than in serum [3]. These properties make this group of drugs very effective against genitourinary tract infections.

We decided to analyze ciprofloxacin and levofloxacin because they are the most commonly used fluoroquinolones in urology [4]. Ciprofloxacin belongs to the second generation of quinolones, in which the fluorine atom is attached to the C6 position of the quinolone scaffold. Such a modification significantly increases drug penetration in bacterial cells, increases DNA gyrase inhibition (>10 fold), and is responsible for a wide range of drug activity [5,6]. Levofloxacin belongs to the third generation of quinolones, it is an isoform of ofloxacin and has a complex ring connected to the oxazinoquinoline core [7]. Levofloxacin shows better activity against Gram-positive bacteria and is less likely to select resistant strains compared to older quinolones [7].

Urinary bladder and prostate cancers are respectively the fourth and the first most common malignancies among males in Europe [8]. Transurethral resection of bladder tumor or radiation/hormone therapy, in the case of prostate cancer, are not effective enough and often lead to relapse and disease progression. In both cases, as a consequence, a radical cystectomy or prostatectomy is required [9,10]. The low effectiveness of available treatment methods has encouraged researchers to work on the new anticancer drug candidates.

Introducing new drugs for clinical application is a long, drawn-out, and costly process. For these reasons, in recent years, fewer and fewer new drugs were registered for clinical use. Drug repositioning (searching for the new target for drugs currently registered for use) allows introducing a medicament for new clinical indications without such difficult procedures [11]. Currently, fluoroquinolones are used mainly as antibacterial agents, but various in vitro studies indicates that this group of drugs is effective against many cancer cells, including bladder and prostate cells [12,13]. Anticancer properties together with the accumulation of quinolones in urine and prostate tissue makes this group of antibiotics promising candidates for application in cancer therapy.

The aim of this study was to compare properties of two fluoroquinolones, ciprofloxacin and levofloxacin, as potential drugs for genitourinary cancer treatment. This is the first study concerning the direct comparison of these two drugs. Very important aspect in testing potential anticancer drugs is their influence on surrounding healthy tissues, which is why in this study, we compared the effect of both drugs between cancer cell lines and their non-malignant counterparts. In previous studies analyzing fluoroquinolones, such a comparison was performed very rarely. Most of the studies were focused on one type of disease or, when analyzing fluoroquinolones modification, on the panel of cancer cell lines. We decide to analyze two malignancies: prostate and bladder cancer. The reason of choosing these conditions was that both drugs are often used in prostate and urinary bladder infections treatment. That is why the repositioning of ciprofloxacin and levofloxacin for these two new clinical indication seems to be reasonable.

## 2. Materials and Methods

### 2.1. Cell Lines

SV-HUC-1 (non-malignant human urothelium), T24 (human bladder cancer), DU-145 (human prostate cancer), and RWPE-1 (non-malignant human prostate epithelium) cell lines were purchased from the American Type Culture Collection (Manassas, VA, USA). T24 and DU-145 were cultured in DMEM/Ham’s F-12 medium (Corning, New York, NY, USA) containing 10% fetal bovine serum (FBS, Pan-Biotech GmbH, Aidenbach, Germany) supplemented with 5 μg/mL of amphotericin B (GE Healthcare Life Science, Chicago, IL, USA), 100 μg/mL of streptomycin, and 100 U/mL of penicillin (GE Healthcare Life Science, Chicago, IL, USA). SV-HUC-1 was cultured in F-12K medium (Corning, New York, NY, USA) with the same supplementation. RWPE-1 was cultured in KBM-Gold™ Keratinocyte Cell Basic Medium (Lonza, Basel, Switzerland) supplemented with a KGM-Gold SingleQuot Kit (Lonza, Basel, Switzerland). All cell lines were grown in a plastic tissue culture 75 cm^2^ T-flasks (Corning, New York, NY, USA) at 37 °C and 5% CO_2_.

### 2.2. Drugs

Ciprofloxacin and Levofloxacin working solutions (25–800 μg/mL; concentrations established on the basis of previous studies) were prepared by dilution of ready to use for intravenous infusions of ciprofloxacin (10 mg/mL, Cipronex, KRKA, Novo Mesto, Slovenia) and levofloxacin (5 mg/mL, Levoxa, Actavis, Reykjavik, Iceland) solutions in media used for cell culture.

### 2.3. MTT Assay

Cell viability after drug treatment was analyzed using MTT assay. For this purpose, cells were seeded in 24-well plates and treated with different concentrations (25, 50, 100, 200, 500, and 800 μg/mL) of ciprofloxacin and levofloxacin for 24 and 48 h. Photographic documentation of cells after incubation with drugs was performed using a light-inverted microscope Leica DMi1 (Leica, Wetzlar, Germany). The morphometric analysis of cells by measuring their area was performed using EPview 1.3 software (Olympus, Tokyo, Japan). Post treatment, MTT solution (1 mg/mL, Millipore Sigma, St. Louis, MO, USA) was added to each well and incubated for 2 h at 37 °C in dark. Formazan crystals were dissolved in dimethyl sulfoxide (DMSO, POCH, Gliwice, Poland), and the absorbance was measured at 570 nm (characteristic length) and 655 nm (reference length) using the Varioskan LUX plate reader (Thermo Fisher Scientific, Waltham, MA, USA). Each experiment was performed at least in triplicate. The results of the MTT assay, collected from all performed repetitions, were used to calculate Lethal Concentrations (LC) causing death of 10%, 50%, and 90% of cells—LC10, LC50, and LC90. The reduction of cell viability was calculated by comparison of the absorbance in drug treated cells to untreated cells as a control. On the basis of the obtained results, logarithmic trend lines were fitted and a curve equation was generated, which allowed for appropriate LC value calculation.

### 2.4. Real-Time Cell Growth Analysis

In order to confirm the calculations after MTT assay, LC10, LC50, and LC90, real-time cell growth analysis using the xCELLigence RTCA DP (ACEA Bioscience, San Diego, CA, USA) was performed. Cells were seeded in an E-Plate 16 (ACEA Bioscience, San Diego, CA, USA) and incubated in appropriate cell culture medium for 24 h. In the next step, ciprofloxacin or levofloxacin in concentrations corresponding to LC values were added. For representation of changes in cell growth, an electrical impedance was measured every 30 min for 24 or 48 h and presented as a cell index (CI). The cell index values can be used to monitor the cell number, viability, morphology, and adhesion degree. A well with medium without cells served as the background, and cells cultured with medium without drugs were used as a control. Each experiment was performed in triplicate.

### 2.5. Cell Cycle Analysis

After the exposition of cells to LC values calculated after 24 h incubation with ciprofloxacin or levofloxacin, cell cycle analysis with a Tali^®^ Cell Cycle Kit (Thermo Fisher Scientific, Waltham, MA, USA) was performed. This method allows for cell cycle phase determination by quantifying the amount of DNA in the cell. Propidium iodide (PI) and RNAse mix were used for DNA staining. Detached cells were fixed in 70% ethanol at −20 °C and kept in these conditions for at least 24 h before staining. After washing from ethanol, cells were stained with PI for 30 min at room temperature. Stained cells were analyzed by flow cytometry with BD FACSCanto II (BD Biosciences, Franklin Lakes, NJ, USA) (Appendix A). The obtained data were analyzed using FlowJo v10 (Becton, Dickinson and Company, Franklin Lakes, NJ, USA). The percentage of cells in the G0/G1, S, and G2/M phases were calculated using a Watson model. Experiment was performed in triplicate.

### 2.6. Apoptosis Detection

The level of apoptosis was assessed based on the quantification of phosphatidylserine (PS) level on the outer cell membrane. After exposition to LC values calculated after 24 h incubation with ciprofloxacin or levofloxacin, cells were detached and stained with the FITC Annexin V Apoptosis Detection Kit II (BD Biosciences, Franklin Lakes, NJ, USA). The staining procedure was performed according to the manufacturer’s instructions. Cells stained with annexin V (AnV) and propidium iodide (PI) were analyzed by flow cytometry with a BD FACSCanto II (BD Biosciences, Franklin Lakes, NJ, USA) (Appendix A). The obtained data were analyzed using BD FACSDiva Software (BD Biosciences, Franklin Lakes, NJ, USA). The experiment was performed in triplicate.

### 2.7. Caspases Detection

The caspase activity was measured using Caspace-Glo^®^3/7 and Caspase-Glo^®^9 (Promega, Walldorf, Germany) assays according to the manufacturer’s protocol. Briefly, cells were seeded in proper density on a 96-well black plate with a clear bottom (Eppendorf, Hamburg, Germany). Cells were allowed to attach to the growth surface for 24 h, and next, both tested drugs in LC50 concentration were added for 24 h incubation. After the preparation of both reagents (combining the substrate with buffer; for Caspase-Glo^®^9 assay, additionally, an MG-132 inhibitor was added), reagents were brought to room temperature. Next, an equal volume of reagent was added directly to cells in culture medium. After that, the plate was placed in a Varioskan LUX plate reader (Thermo Fisher Scientific, Waltham, MA, USA), shaken for 30 s at 300 rpm, and incubated in the dark for 30 min, after which the luminescence signal was measured. Experiment was performed in triplicate.

### 2.8. Gene Expression Analysis

The expression of *BAX, BCL2, TOP2A, TOP2B, CDKN1A,* and *TP53* genes was analyzed by real-time PCR. The RNA was extracted using the RNeasy Mini Kit according to the manufacturer’s protocol (Qiagen, Hilden, Germany). cDNA was synthesized from 500 ng of total RNA with the use of a Transcriptor High Fidelity cDNA Synthesis System (Roche Diagnostics, Basel, Switzerland). Quantitative real-time PCR was performed using PrimePCR™ primers (Bio-Rad, Hercules, CA, USA) and a LightCycler 480 SYBR Green I Master (Roche Diagnostics, Basel, Switzerland). NormFinder was used to analyze the expression stability of reference genes and to select the combination of the most stable genes (*SDHA* and *TBP*). The Roche LightCycler 480 software 1.5 (Roche Diagnostics, Basel, Switzerland) was used to perform advanced relative quantification analysis.

### 2.9. 3D Cell Culture

Spheroids were generated from T24 and SV-HUC-1 cell lines. For this purpose, cells were seeded at a density of 25,000 cells per well in low-attachment 96-well U-shaped Nunclon^TM^Sphera^TM^ plates (Thermo Fisher Scientific, Waltham, MA, USA). Spheroids were grown for 3 to 4 days, after which analysis was performed. For analysis of cell viability, a CellTiter-Glo^®^ 3D assay was used (Promega, Walldorf, Germany). Both tested drugs in calculated LC concentrations were added to the generated spheroids and incubated for 24 and 48 h. The day before analysis, the reagent was thawed at 4 °C; directly before use, it was placed in a water bath (22 °C) for 30 min. Spheroids were transferred to a low-attachment, 96-well, U-shaped PrimeSurface^®^ white plate (PHC Europe, Utrecht, The Netherland), and next, an equal volume of reagent was added directly to cells in the culture medium. After that, the plate was placed in a Varioskan LUX plate reader (Thermo Fisher Scientific, Waltham, MA, USA), shaken for 5 min at 420 rpm, and incubated in the dark for 30 min, after which the luminescence signal was measured. In the case of caspase activity measurement, a Caspase-Glo^®^3/7 3D assay (Promega, Walldorf, Germany) was used using similar protocol. Before use, the substrate was combined with buffer, and the obtained reagent was brought to room temperature. The difference in protocol compared to the CellTiter-Glo^®^ 3D assay was the shaking parameters (30 s at 500 rpm). The experiment was performed in triplicate. Photographic documentation of spheroids after incubation with drugs was performed using a light-inverted microscope Leica DMi1 (Leica, Wetzlar, Germany). Morphometric analysis of spheroids by measuring their area was performed using EPview 1.3 software (Olympus, Tokyo, Japan).

### 2.10. Statistical Analysis

Each experiment was performed at least in triplicate. The average cell viability was expressed as a percentage relative to the control. All data were presented as means ± SD. Normal distribution of data was analyzed using the Shapiro–Wilk test. Parametric analysis was performed with one-way ANOVA with Tukey post hoc (for cell viability) or two-way ANOVA with Sidak post hoc (for grouped analysis). Non-parametric analysis was performed using the Kruskal–Wallis test. In order to compare the cytotoxicity profiles of both tested drugs on all tested cell lines, Pearson comparison was performed GraphPad Software 8.4 (GraphPad Software, San Diego, CA, USA).

## 3. Results

### 3.1. Ciprofloxacin

#### 3.1.1. Cell Morphology

Morphological changes in the bladder cell lines (both non-malignant SV-HUC-1 and cancer T24) including cell shrinkage, rounding, and detachment were visible after ciprofloxacin treatment, especially in higher concentrations (Figure 1A). Twenty-four hour incubation of SV-HUC-1 and T24 with a low concentration of ciprofloxacin (100 µg/mL) led to cell shrinkage and their detachment (Figure 1A). In higher ciprofloxacin concentrations (200 and 800 µg/mL), only a small number of cells remained attached. Crystals of ciprofloxacin started to appear in culture in 100 µg/mL (SV-HUC-1) or in the highest concentration (800 µg/mL, T24). Morphometric analysis showed in most cases a lack of changes in the cells’ area in lower drug concentrations compared to control. Cells area reduction was observed in higher (200 and 800 µg/mL) drug concentrations (Figure 1B).

Morphological changes of prostate cell lines (RWPE-1 and DU-145) were also observed after ciprofloxacin treatment, especially in higher concentrations. Cell shrinking, rounding, and detachment that led to a decrease in cell number with increasing drug concentration were visible. Changes in cell morphology were more visible in the case of cancer DU-145 cells, in which detachment and shape change, into a more rounded shape, were observed already in 100 µg/mL concentration, especially after 48 h incubation (Figure 2A). Morphometric analysis showed a lack of changes in the cells’ area in lower drug concentrations compared to control. The cell area reduction was observed in higher (200 and 800 µg/mL) drug concentrations (Figure 2B).

#### 3.1.2. Cell Viability

Ciprofloxacin acted cytotoxic against all of the tested bladder and prostate cell lines in a dose and time-dependent manner. A lack of significance was observed in the case of 25 µg/mL and 50 µg/mL for SV-HUC-1 and 25 µg/mL for T24 (Appendix A). Ciprofloxacin was more effective against bladder cancer cells (Figure 3A,B). A lack of significance was observed only at the highest concentration of ciprofloxacin (500 and 800 µg/mL after 24 h and 800 µg/mL after 48 h). Such a relationship was not observed in the case of prostate cell lines; differences were noticed between single concentrations: once in favor of non-malignant and the second time in favor of the cancer cell line.

MTT assay enabled for lethal concentration (LC) calculation causing death of 10%, 50%, and 90% cells in vitro (respectively LC10, LC50, and LC90). These values were used in further analysis (Table 1 and Appendix A). Real-time analysis of cell growth with the use of calculated LC values confirmed the results obtained with MTT assay (Figure 4).

#### 3.1.3. Changes in Cell Cycle

The results of cell cycle analysis revealed that ciprofloxacin caused a rise of the cell number in the S phase, especially after treatment with LC90 concentrations. In the case of bladder cell lines, this effect was especially visible in cancer cells, while in non-malignant urothelium, only slight changes were observed. However, statistically significant results were observed only in the case of the T24 cell line treated with ciprofloxacin (decrease in cell number in the G1/G0 phase compared to control after LC90 treatment). The effect of ciprofloxacin on prostate cell lines was the opposite: larger differences were observed in the non-malignant epithelium compared to the cancer prostate epithelium. In this case, a lack of statistically differences were observed (Appendix A and Figure 5).

#### 3.1.4. Induction of Apoptosis

Flow cytometry analysis revealed an increased number of AnV+/PI+ cells after the incubation of tested cells with ciprofloxacin in LC90 concentration (*p* < 0.005). The obtained results were similar to cell cycle analysis; in bladder cell lines, an increased number of AnV+/PI+ cells was especially visible in cancer cells, while in prostate cell lines, the situation was the opposite (Figure 5). The activation of caspase 3/7 was observed in the case of ciprofloxacin in SV-HUC-1 and T24 cell lines. In the case of caspase 9, a decrease in luminescence signal was observed in all tested cell lines (Appendix A).

#### 3.1.5. Molecular Analysis

Both *BAX* and *BCL2* genes were overexpressed in the T24 cell line following ciprofloxacin treatment, in the case of the SV-HUC-1 line, only the *BAX* gene was overexpressed in LC90 concentration. In the case of prostate cell lines, an increase in *BAX* expression was observed in LC10 and LC50 concentration, while in LC90, a decrease was noticed; no significant changes were observed in the *BCL2* gene. In all tested cell lines, a decrease in *TOP2A* and *TOP2B* genes were observed, especially in higher tested concentrations. An increased expression of the *TP53* gene was observed after treatment with LC50 concentration on the SV-HUC-1 cell line; in the case of the T24 cell line, expression of *TP53* was not observed. In the case of prostate cell lines, an increased expression of the *TP53* gene was observed in LC50-treated cancer cells, while in both cell lines, in the highest tested concentrations, a decrease in gene expression was observed (in the case of RWPE-1, expression of *TP53* was not observed). In both bladder cell lines, a decrease in the *CDKN1* gene was observed, while in the case of the prostate cell line, a decrease was observed in LC90 in RWPE-1 and LC10 and LC90 in DU-145 cell lines. In the case of the DU-145 cell line, an increase in *CDKN1* expression was observed after treatment with LC50 concentration (Figure 6).

#### 3.1.6. Effect of Ciprofloxacin on 3D Culture

Viability assay showed that only the calculated LC90 concentration was effective in reducing T24 cell viability in 3D culture. In the case of the SV-HUC-1 cell line, all calculated LC concentrations were not effective. An increase in caspase 3/7 activity was observed in the case of the T24 cell line, which was especially pronounced after incubation with LC90 concentration for 48 h. In the non-malignant uroepithelial cell line, a small increase in caspase activity was observed in LC50 and LC90 concentrations after 48 h incubation (Figure 7). Morphological analysis of spheroids showed a lack of changes in LC10 and LC50 concentrations, while in LC90, an increase in spheroids diameter (by 30–40%) was observed (Figure 8).

### 3.2. Levofloxacin

#### 3.2.1. Cell Morphology

In the case of levofloxacin, the results were similar; however, due to lower cytotoxic activity, the number of cells was higher compared to the cells cultured in the same concentration of ciprofloxacin. In the case of levofloxacin, no crystal formation in culture was observed (Figure 9A). Morphometric analysis showed a lack of changes in the area of SV-HUC-1 cells following levofloxacin treatment, while in the case of the T24 cell line, an increase in cell area, compared to the control, was observed (Figure 9B).

As in the case of bladder cell lines, levofloxacin caused a reduction in RWPE-1 and DU-145 cell number with increasing drug concentration. The changes in cell morphology were less severe compared to ciprofloxacin. No crystal formation in culture medium was observed (Figure 10A). Morphometric analysis showed a lack of changes in the cells area in lower drug concentrations compared to control. Cell area reduction, in most cases (except DU-145 after 48 h incubation with drug), was observed in higher (200 and 800 µg/mL) levofloxacin concentrations (Figure 10B).

#### 3.2.2. Cell Viability

Levofloxacin acted cytotoxic against all tested bladder and prostate cell lines in a dose- and time-dependent manner. Comparison between 24 and 48 h incubation with drugs showed a lack of statistical significance in the range between 25 and 200 µg/mL in the case of levofloxacin on RWPE-1 and 25 µg/mL concentration on DU-145 (Appendix A). Levofloxacin was more effective against bladder cancer cells (Figure 3A,B). A lack of significance was observed only at the low concentrations of levofloxacin (25 and 50 µg/mL) after 48 h. Such a relationship was not observed in the case of prostate cell lines. Differences were noticed between single concentrations, cell viability was higher in non-malignant cell lines or cancer cells without any noticeable pattern (Figure 3A,B).

MTT assay enabled lethal concentration (LC) calculation causing the death of 10%, 50%, and 90% cells in vitro (respectively LC10, LC50, and LC90). These values were used in further analysis (Table 1 and Appendix A). Real-time analysis of cell growth with the use of calculated LC values confirmed the results obtained with MTT assay (Figure 11).

#### 3.2.3. Changes in Cell Cycle

The results of cell cycle analysis revealed that levofloxacin caused a rise of cells number in the S phase, especially after treatment with LC90 concentrations. Levofloxacin increased the number of cells in the S phase in both bladder cell lines; in bladder cancer cells, an increased number of cells in the G2/M phase was also observed. In the case of prostate cell lines, a similar increase in cell number in the S phase was observed; this effect was more pronounced in cancer cells, but the results compared to control were not statistically significant (Appendix A and Figure 5).

#### 3.2.4. Induction of Apoptosis

Flow cytometry analysis revealed an increased number of AnV/PI+ cells after the incubation of tested cells with levofloxacin in LC90 concentration (*p* < 0.005). Levofloxacin induced an increase in the number of AnV+/PI+ cells in all tested cell lines (*p* < 0.005); in the case of cancer cells, this effect was more visible (Figure 5). The activation of caspase 3/7 was observed in the case of levofloxacin in T24 cell lines. In the case of caspase 9, a decrease in luminescence signal was observed in all tested cell lines (Appendix A).

#### 3.2.5. Molecular Analysis

The *BAX* gene was overexpressed in all tested cell lines after levofloxacin treatment in LC90 and in most cases downregulated in LC50 concentration. The *BCL2* gene was upregulated in the SV-HUC-1 cell line and downregulated in the T24 cell line treated with LC90 concentration. In the case of the prostate cell lines, an upregulation of the *BCL2* gene was observed in the highest tested concertation, which was more pronounced in cancer cell lines. At LC50 concentration, an upregulation of the *BCL2* gene in RWPE-1 and downregulation of the *BCL2* gene in DU-145 was observed. In almost all the tested cell lines, a decrease in the *TOP2A* and *TOP2B* genes expression were observed, especially in higher tested concentrations. Only in the case of the RWPE-1 cell line we did observe an upregulation of *TOP2A* in LC50 concentration. A decreased expression of the *TP53* gene was observed after treatment of LC10 concentration on the SV-HUC-1 cell line; in the case of the T24 cell line, expression of *TP53* was not observed. In the case of the DU-145 cell line, no significant differences in *TP53* gene expression were observed, while in RWPE-1 cells, a decrease in LC50 and increase in LC90 drug concentrations was noticed. In both bladder and prostate non-malignant cell lines, an increase in the *CDKN1* gene was observed in LC50 (SV-HUC-1) and LC50 and LC90 (RWPE-1). In the case of cancer cell lines, a decrease in *CDKN1* expression was observed in the T24 cell line, while in the case of DU-145, a lack of significant changes was noticed (Figure 12).

#### 3.2.6. Effect of Levofloxacin on 3D Culture

Viability assay showed that only the LC90 concentration calculated for levofloxacin was effective in reducing T24 cell viability in 3D culture. In the case of SV-HUC-1, a reduction of cell viability was also observed in LC90 concentration; however, compared to the T24 cell line, the effect of levofloxacin was weaker. An increase in caspase 3/7 activity was observed in the case of both tested cell lines, which was especially pronounced after incubation with LC90 concentration for 48 h in the T24 cell line. In the non-malignant uroepithelial cell line, the increase in caspase activity was weaker (Figure 7). A morphological analysis of spheroids showed a lack of changes in most of the analyzed spheroids with the exception of the T24 cell line after 24 h of incubation with LC90 concentration, in which about a 15% increase of the spheroids area was observed (Figure 13).

## 4. Discussion

Despite many therapeutic possibilities, including surgical methods accompanied by chemotherapy, radiotherapy, and immunotherapy, many deaths related to prostate and bladder cancers are still observed. The reported number of deaths caused by these types of cancers constitutes 28% of newly diagnosed prostate and 40% of bladder cases [5]. This fact can be caused by the insufficient efficiency of treatments supplemental to the surgical methods. Therefore, the development of supportive treatment at the early stage of disease is necessary to decrease the number of relapses. The production of a new drug and its introduction into the market is a very difficult, time-consuming (13–15 years), and cost-consuming (≈$2.6 billion) process. By 2016, about 22 new drugs were approved for use, which is a 2.5-fold lower number compared to results from two decades ago [11]. That is why drug repositioning has been gaining importance over the last years. Fluoroquinolones have large potential in this field. Their potential application as anticancer agents in prostate and bladder treatment can be possible from two important reasons: an accumulation in higher concentration in urine and the prostate gland than in serum and confirmed anticancer properties in vitro.

The concentration of ciprofloxacin achievable in urine can range from 205 to 1087 µg/mL depending on the dose of drug (respectively 250–750 mg) [14,15]. Taking into account the results obtained in this study, only LC90 obtained after 24 h incubation with this drug is not achievable in urine. In the case of levofloxacin, the administration of a 500 mg dose resulted in a 406 µg/mL concentration in urine, and one dose of 750 mg resulted in a 620 µg/mL concentration in urine. Interestingly, this value has not increased after the administration of one 1000 mg dose (536 µg/mL) [16,17]. The results from our study indicated that in this case, LC10 and LC50 calculated for a bladder cancer cell line can be obtained in urine. Both fluoroquinolones can be administered in high doses for longer periods of time (750 mg for more than 7 days), which will increase concentration in urine, allowing the achievement of almost all the LC values calculated in this study [18,19].

Concentrations of fluoroquinolones in prostate tissue are 2.45-fold higher in the case of ciprofloxacin and 2.96-fold higher in the case of levofloxacin compared to serum. However, the penetration of drugs in such a way enables to obtain concentrations of 5.81 µg/g and 4.49 µg/g after administration of respectively 200 mg and 1000 mg ciprofloxacin and 20.8 µg/g after 500 mg of levofloxacin [20,21,22,23]. The lowest LC calculated in this study was 15 µg/mL for the prostate cancer cell line after 48 h incubation, and only this concentration can be achieved in the prostate gland.

Ciprofloxacin was more cytotoxic than levofloxacin for all tested cell lines. The highest differences were observed in the case of the bladder cancer cell line, in which a lack of difference between ciprofloxacin and levofloxacin cytotoxicity was observed only in the case of the highest concentration after 48 h incubation (Figure 3A). In the case of non-malignant urothelium, no differences were observed in the lowest tested concentration in both incubation times (Appendix A). Significant differences in the drugs’ activity against the prostate cancer cell line were observed in the highest (200–800 µg/mL) tested concentrations after 24 h of incubation; after 48 h, a lack of significance was observed only in the concentration of 50 µg/mL (Figure 3B). In the case of non-malignant prostate epithelium, after 24 h incubation, a significant difference was observed only in 500 µg/mL. After 48 h, the results were the opposite: only in the highest concentration a difference was not observed (Appendix A). The observed results were confirmed by a Pearson comparison of cytotoxic profiles. Correlation between the prostate cell lines treated with both tested drugs were very high in all combinations, reaching values above 0.9. This analysis showed a very high similarity of the obtained cytotoxic profiles between both tested drugs and between non-malignant and cancer prostate cell lines. In the case of bladder cell lines, the correlation index was also high, but the obtained values compared to prostate cell lines were also under 0.9, which suggest higher differences between the analyzed cytotoxic profiles. These results are consistent with the results analyzed by a two-way ANOVA test, indicating the advantage of ciprofloxacin over levofloxacin against a bladder cancer cell line (Figure 3C). Both tested drugs differ in molar weight (331.346 g/mol for ciprofloxacin and 361.369 g/mol for levofloxacin). Despite these differences, the calculation of LC values into micromolar units also showed the advantage of ciprofloxacin; lower values were calculated for levofloxacin only in the case of LC10 for the DU145 cell line after 24 h and 48 h incubation (Table 1).

Crystal formation in higher concentrations of ciprofloxacin was observed in all tested cell lines. In our opinion, crystal formation did not significantly affect the concentration of ciprofloxacin in cell culture. In the case of levofloxacin, crystals did not appear, and cell viability was higher compared to these same concentrations of ciprofloxacin. Differences in crystal formation between cell lines (higher number in SV-HUC-1 and DU-145) were caused probably by the different compositions of culture medium. Fluroquinolones dissolve better in acidic pH. Despite that fact, that pH of culture medium is generally neutral; it changes into more acidic, during the culture, when cells grow and consume nutrients. High drug concentrations do not allow for proper cell proliferation, which is why the pH of the medium is not lowered. This property of ciprofloxacin can be dangerous; application of this drug can cause crystal nephropathy in patients with alkaline urine [24]. Such a situation in the case of levofloxacin occurs very rarely [25].

Both ciprofloxacin and levofloxacin were examined earlier on bladder cancer cell lines. Ciprofloxacin was the most examined fluoroquinolone in this field; it was tested on nine different bladder cancer cell lines (EJ, T24, J82, TCCSUP, HTB9, HT1197, HT1376, MBT-2, BC-587) [26,27,28,29,30,31,32,33]. The LC50 values obtained in our study are similar to the results from the study of Kamat et al. (100 µg/mL after 24 h and 50 µg/mL after 48 h); in another study, the LC50 calculated after 24 h incubation with ciprofloxacin was two times higher (160 µg/mL) [28,30,31]. In all studies, growth inhibition with increasing drug concentration was observed. Similar to our observations, an inhibition of cell cycle in the S phase was observed, which was indicated by the modulation of cyclin B and E, CDK2, and a decrease in P21 protein expression [29,30]. In addition, an increased number of apoptotic cells were noticed after ciprofloxacin treatment [29,32]. Molecular analysis indicated an increased expression of BAX protein [29]. In our study, *CDKN1A* expression (coding p21 protein) was also decreased together with the upregulation of the pro-apoptotic gene *BAX*, which indicates an inhibition of cell cycle in the S phase and induction of apoptosis. However, an increase in *BCL2* expression was also noticed (Figure 6A). We additionally performed an analysis of *TP53* gene expression and observed an increased expression in the SV-HUC-1 cell line treated with an LC50 concentration of ciprofloxacin, which can suggest the activation of this protein in response to the cellular damage caused by this drug. In the case of the T24 cell line, expression of *TP53* was not observed, which was caused probably by the mutation of this gene in this cell line [34]. Levofloxacin was tested on two bladder cancer cell lines (T24 and BOY). It is hard to compare the obtained cytotoxic results, because in the work of Yamakuchi et al., analysis was performed after 96 h incubation with the drug [35]. In this study, a reduction of DNA content and lack of changes in cell cycle were observed, while in our study, an increase in cell number both in the S and G2/M phases was observed (*p* > 0.05). The influence of levofloxacin at the molecular level has not been analyzed so far. Similarly, as in the case of ciprofloxacin, *BAX* and *BCL2* genes were upregulated except in the T24 cell line, in which the *BCL2* gene was downregulated. *TP53* and *CDKN1A* genes were downregulated except in the SV-HUC-1 cell line in LC50 concentration, in which the *CDKN1A* gene was overexpressed. The *TP53* gene was not detected in the T24 cell line, which is similar to the case of ciprofloxacin, while in the case of non-malignant uroepithelium, decreased expression was observed (Figure 12A).

Only ciprofloxacin was examined against prostate cancer cells earlier [36,37,38,39]. Two cell lines were analyzed before (PC3 and LNCaP), while in our study, we used the DU-145 cell line, which is why a direct comparison of results is not possible. LC50 values calculated after 24 h incubation with drug were 254 µg/mL and 172 µg/mL, and after 48 h, they were 79 µg/mL and 80 µg/mL for PC3 and LNCaP, respectively. Our results are more comparable to LNCaP cells (165 µg/mL after 24 h and 81 µg/mL after 48 h). The inhibition of cell growth together with sensitization to doxorubicin, docetaxel, mitoxantrone, etoposide, and vinblastine was noticed [36,37,38]. Ciprofloxacin caused an alteration in Bax/Bcl-2 ratio, downregulation of p21^WAF1^, and NF-κB inactivation [38]. In a study performed on seven different cancer cell lines, including PC3, an increased number of cells in the S and G2/M phases and induction of apoptosis/necrosis were visible, which is consistent with the results obtained in this work [39]. Our molecular analysis showed increased *BAX* expression with a lack of significant changes in the *BCL2* gene in LC10 and LC50 concentrations, which suggests apoptosis induction. In LC90, a lack of changes compared to control or a decrease in *BAX* expression was observed (in RWPE-1 and DU-145 respectively), which can indicate the advantage of the necrotic pathway. *CDKN1A* and *TP53* genes were downregulated in L50 and LC90 concentrations except for DU-145, in which an overexpression of the *TP53* gene and *CDKN1A* gene was observed in LC50 concentration, which could suggest that in this case, the pro-survival pathways were involved (Figure 6B). Analysis of the effect of levofloxacin on prostate cell lines has not been performed so far. Our results showed a toxic effect of this drug against DU-145 and RWPE-1 cell lines, which was weaker compared to ciprofloxacin (Figure 3B,C). Molecular analysis revealed that *BAX* and *BCL2* increased expression in the highest tested concentration. In the case of protein engaged in cell cycle regulation, an increased expression of *CDKN1A* and *TP53* genes were observed in the non-malignant prostate cell line without changes in cancer cells (Figure 12B).

Summarizing the molecular analysis and caspases activity, the mechanism of tested fluoroquinolones was dependent on the type of cell line and tested drug concentration. Generally, in lower concentrations, we observed pro-survival pathways activation, while in the highest (LC90) concentration, the *TP53* and *CDKN1A* genes were downregulated. *BAX* and *BCL2* genes activation suggest the participation of the mitochondrial pathway in cell death. On the other hand, a lack of caspase 9 activation and activation of caspase 3/7 only in urothelial cells were observed. This phenomenon could be explained by flow cytometric analysis, in which mainly AnV+/PI+ cells were detected (which are late apoptotic/necrotic/dead cells); that is why an active form of caspases could not be detected in analyzed cells. Cell death can be also induced by other mechanisms such as the TNF signaling pathway, which could lead to caspase 3/7 activation. Other possible mechanisms are the activation of caspase 3/7 omitting caspase 9 by *DIABLO*, *ARTS,* and *HTRA2* activation. Apoptosis can be also induced independently for caspase by *ENDO-G* and *AIF* activation [40]. Taking into account previous results performed using fluoroquinolones, the first proposed hypothesis is the most likely.

Our results showed that both non-malignant and cancer prostate cell lines were comparably sensitive to both drugs in 2D culture. These results suggest that fluoroquinolones in high concentrations can damage heathy tissue. Available data showed that ciprofloxacin can be administered in a high dose (1000 mg) without any drug-related adverse events [21]. However, the concentration of both drugs achievable in prostate tissue is relatively low [21], which is why it is not possible to evaluate how prostate cells will respond to higher doses of fluoroquinolones, for example after intratissue injection. In this study, we used pharmaceutical products that contain several auxiliary substances such as lactic acid, hydrochloric acid, edetate sodium, and water for injection in the case of ciprofloxacin and sodium chloride, 5N hydrochloric acid, and water for injections in the case of levofloxacin. These substances could have influenced cell growth; however, the aim of our study was to analyze the finished and registered product in order to repositioning it for other indications.

As we mentioned before, the mechanism of fluoroquinolones action is topoisomerase inhibition. The most preferred targets for fluoroquinolones are bacterial gyrase and topoisomerase IV. This group of drugs is safe for clinical application because their effectiveness against eucaryotic topoisomerases is much smaller compared to procaryotic counterparts (1000-fold). This effect is probably caused by a lack of serine and acidic amino acid residues in topoisomerase molecules, which in procaryotic cells are responsible for interaction of the drug with the enzyme [41]. That is why the anticancer properties of fluoroquinolones that were confirmed also in our study can be the effect of different mechanisms, which is not clear so far, and this issue is worth further analysis. Our analysis of *TOP2A* and *TOPO2B* genes expression, two isoforms of eucaryotic topoisomerase II (α and β), showed their downregulation in almost all tested cell lines (beside levofloxacin in the RWPE-1 cell line on the *TOP2A* gene) by both fluoroquinolones. To our knowledge, this is the first study confirming the inhibition of topoisomerase II by fluoroquinolones in human cancer and non-malignant cell lines (Figure 6 and Figure 12). These results can indicate a closer mechanism of action of fluoroquinolones on cancer cells, indicating that topoisomerase inhibition plays a key role in this process.

The advantage of our study is its comparison of the cytotoxic effect of both drugs on cancer and non-malignant bladder and prostate epithelial cells. Such a comparison has not been performed before, which is why in our study, we proved that both tested fluoroquinolones were more effective against bladder cancer cell lines (Figure 3A,C). This effect was not observed in the case of prostate cell lines (Figure 3B,C). Additionally, this is the first study comparing the effects of both drugs. Ciprofloxacin and levofloxacin are the most prescribed fluoroquinolones in urology [4]. Taking into account the results obtained for cancer cell lines, ciprofloxacin is more effective compared to levofloxacin. This effect is more pronounced in the case of bladder cancer cells (Figure 3). The promising results obtained for the bladder cell lines encouraged us to perform the first experiments investigating the influence of ciprofloxacin and levofloxacin on 3D culture. The generated spheroids were resistant to calculated for 2D culture LC10 and LC50 concentrations; a cytotoxic effect was visible only in the highest tested concentrations (Figure 7). It is interesting that even in these concentrations, spheroids kept their circular shape and only an increase in spheroids diameter, which resulted from loosening their structure, was noticed (Figure 8 and Figure 13). The reduction of viability was accompanied by an increase in caspase 3/7 activity (Figure 7). There was another interesting result for the T24 cell line after 24 h incubation treated with levofloxacin; in that case, an increase in caspases activity was observed in LC50 concentration and in the case of LC90, a decrease in activity was observed. These results could be caused by the high number of late apoptotic/necrotic/dead cells (AnV+/PI+), as observed in flow cytometric analysis of 2D cultured cells (Figure 5). Caspases may no longer be active in late apoptosis. The results obtained on 3D cultures more clearly showed the advantage of both tested fluoroquinolones against cancer compared with non-malignant cells. Especially, the lack of cytotoxic effect of ciprofloxacin on SV-HUC-1 cells, which was related with a lack of (24 h) or small amount of (48 h) caspases activation, indicated the high therapeutic potential of this drug (Figure 7).

The results obtained in this study indicate that ciprofloxacin and levofloxacin can be use as potential drugs, especially in the treatment of bladder cancer. An ideal indication for the utilization of this drug is condition after TURBT. This surgical procedure is related with a high percentage of relapses, probably by implantation of the remaining cancer cells after resection of the bladder tumor [42,43]. The application of fluoroquinolones directly after the end of the procedure, as an intravesical therapy, and orally for several days after surgery, can eradicate the remaining cancer cells and decrease the number of relapses. The treatment of prostate cancer with fluoroquinolones is less promising, mainly because of the significantly lower concentrations achievable in prostate tissue compared to urine. In order to achieve LC50 and LC90 values in prostate tissue, the delivery of drugs directly into this organ would be needed, which is a much more invasive method than intravesical bladder therapy. Both levofloxacin and ciprofloxacin are used in the treatment of chronic prostatitis [44]. A chronic inflammation process can lead to cancer development; that is why the use of fluoroquinolones allows curing almost all forms of prostatitis and reduces the possibility of cancer development (“two-hit hypothesis”) [45].

The use of both fluoroquinolones in cancer treatment will require a prolonged use of these drugs. The long-term use of ciprofloxacin and levofloxacin is well tolerated, and they are generally considered as safe and efficacious antimicrobial agents [4]. Unfortunately, these drugs were extensively prescribed by some representatives, which contributed to bacterial resistance development [46,47]. Additionally, the use of this group of drugs can lead to the development of side effects such as diarrhea and vomiting, effects on tendons, joints, muscles, and nerves, retinal detachment, aortic aneurysm, and a variety of central nervous system disturbances [48]. That is why, in recent years, organizations such as the Food and Drug Administration (FDA), European Medicines Agency (EMA), and European Association of Urology (EAU) recommended a limited use of fluoroquinolones. However, such long-lasting and disabling side effects appear very rarely [45]. Moreover, compared to commonly used chemotherapeutics such as cisplatin, the side effects, including ototoxicity, neurotoxicity, and nephrotoxicity, are more severe, leading to accelerated aging and the possibility of secondary malignancies [49].

Properties of ciprofloxacin and levofloxacin create opportunities for their use as anticancer agents. They have confirmed in vitro anticancer properties and can achieve higher concentrations in different organs than in serum after oral and intravenous administration. Our results suggest that both tested drugs can be potentially used in supportive therapy of bladder cancer treatment. From both tested fluoroquinolones, ciprofloxacin has better cytotoxic properties. Taking into account the low costs of such therapy, fluoroquinolones seem to be ideal candidates for repositioning into bladder cancer therapeutics.

## Figures and Tables

**Figure 1 ijms-22-11970-f001:**
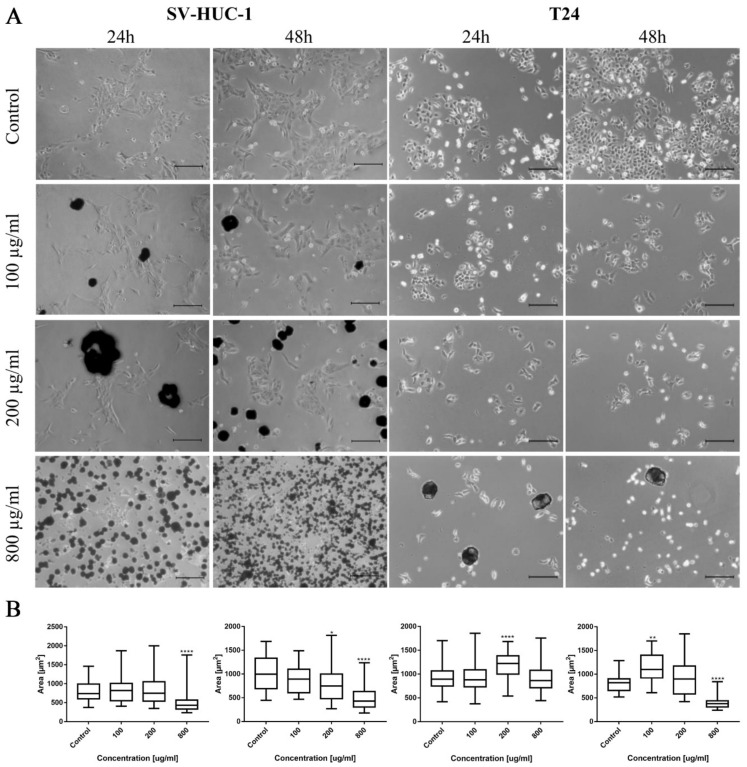
Bladder cell lines characteristic after ciprofloxacin treatment. (**A**)—Morphology of bladder cell lines after ciprofloxacin treatment. Cell shrinkage, rounding, and detachment were visible after ciprofloxacin treatment, especially in higher concentrations. Ciprofloxacin crystal formation was visible in concentration 100 µg/mL for SV-HUC-1 and 800 µg/mL for T24, using an inverted light microscope (bar = 200 µm). (**B**)—Morphometric analysis of cells area. SV-HUC-1—non-malignant human urothelium; T24—human bladder cancer; statistically significant results compared to control were presented: *—*p* ≤ 0.05; **—*p* ≤ 0.01; ****—*p* ≤ 0.0001.

**Figure 2 ijms-22-11970-f002:**
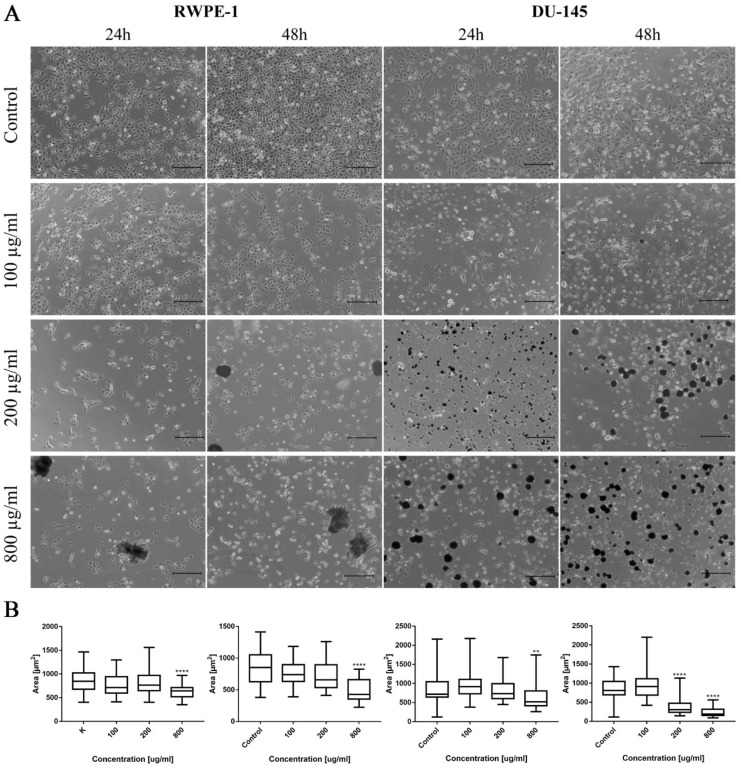
Prostate cell lines characteristic after ciprofloxacin treatment. (**A**)—Morphology of prostate cell lines after ciprofloxacin treatment. Cell shrinkage, rounding, and detachment were visible after ciprofloxacin treatment, especially in higher concentrations. Ciprofloxacin crystal formation was visible in a concentration of 200 µg/mL for RWPE-1 DU-145, using an inverted light microscope (bar = 200 µm). (**B**)—Morphometric analysis of cells area. DU-145—human prostate cancer; RWPE-1—non-malignant human prostate epithelium; statistically significant results compared to control were presented: **—*p* ≤ 0.01; ****—*p* ≤ 0.0001.

**Figure 3 ijms-22-11970-f003:**
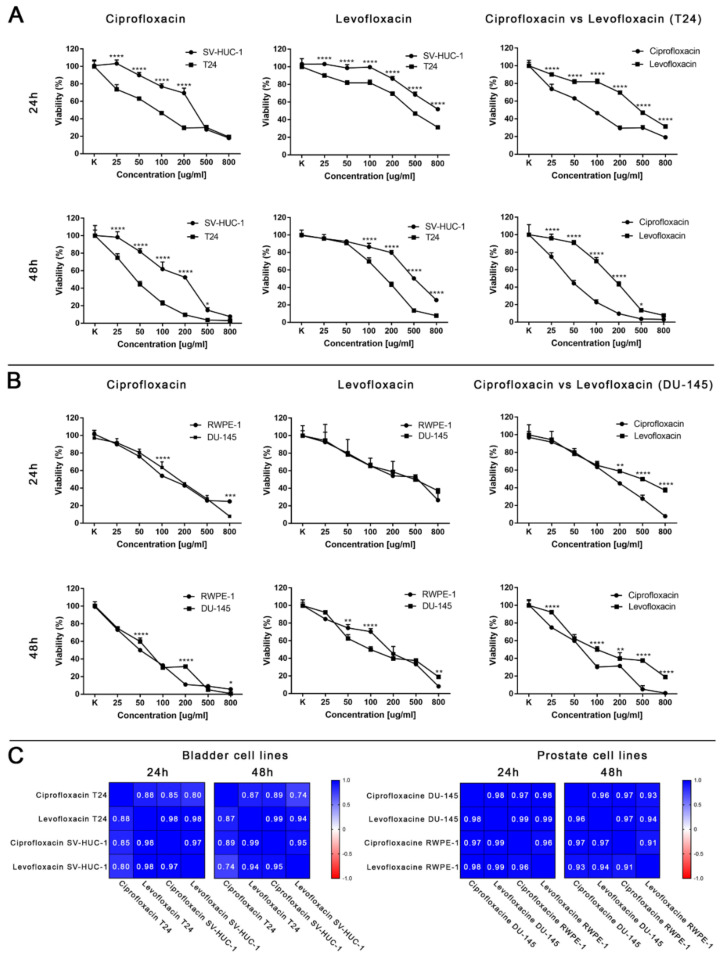
Cytotoxic properties of ciprofloxacin and levofloxacin. (**A**)—Effect of both drugs on bladder cell lines. (**B**)—Effect of both drugs on prostate cell lines. Significance was presented between two tested populations. (**C**)—Pearson comparison of cytotoxic profiles of both drugs on bladder and prostate cell lines. Results obtained using MTT assay after 24 and 48 h incubation with both drugs. Ciprofloxacin and levofloxacin were more effective against cancer cell lines. Comparison of both drugs on the bladder cancer cell line showed the advantage of ciprofloxacin. SV-HUC-1—non-malignant human urothelium; T24—human bladder cancer; DU-145—human prostate cancer; RWPE-1—non-malignant human prostate epithelium; *—*p* ≤ 0.05; **—*p* ≤ 0.01; ****—*p* ≤ 0.0001.

**Figure 4 ijms-22-11970-f004:**
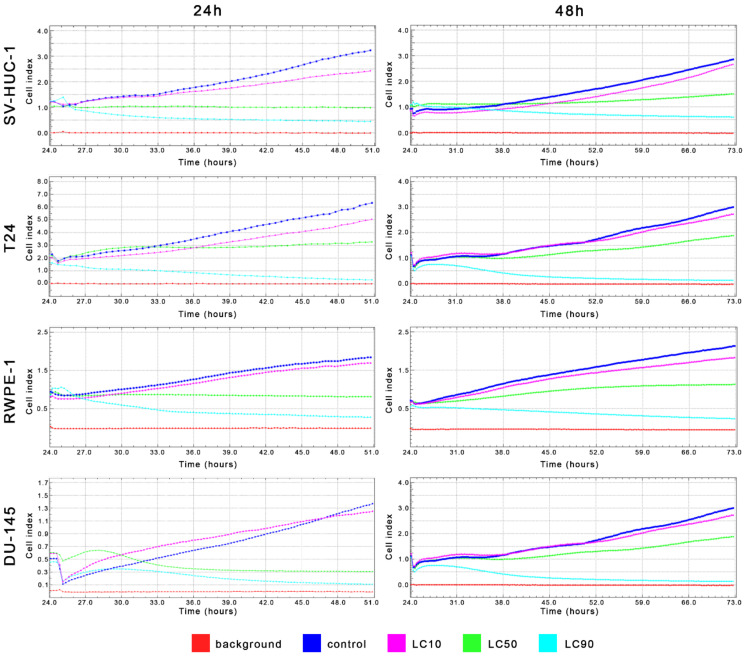
Real-time cell growth analysis of bladder and prostate cell lines after ciprofloxacin treatment. Data were measured every 30 min for 24 or 48 h. Obtained results confirmed that LC values calculated using MTT assay actually cause a reduction of cells viability by 10%, 50%, and 90%. SV-HUC-1—non-malignant human urothelium; T24—human bladder cancer; DU-145—human prostate cancer; RWPE-1—non-malignant human prostate epithelium.

**Figure 5 ijms-22-11970-f005:**
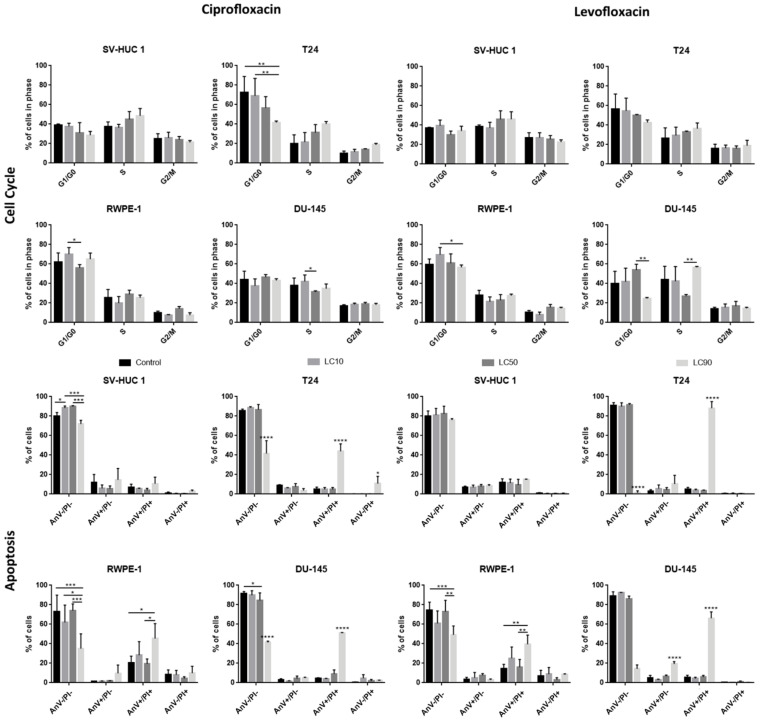
Cell cycle and apoptosis analysis. Both drugs caused a rise of cells number in the S phase, especially after treatment with LC90 concentrations. After levofloxacin treatment in bladder cancer cells, an increased number of cells in the G2/M phase was also observed. Apoptosis analysis revealed an increased number of late apoptotic/necrotic/dead (AnV+/PI+) cells after incubation of tested cells with both fluoroquinolones in LC90 concentration. AnV-/PI-: alive; AnV+/PI-: probably early apoptotic; AnV+/PI+: dead/late apoptotic/necrotic; AnV-/PI+: membrane-free nuclei, where with a decrease in PI signal, the degradation of genetic material is observed; SV-HUC-1—non-malignant human urothelium; T24—human bladder cancer; DU-145—human prostate cancer; RWPE-1—non-malignant human prostate epithelium; *—*p* ≤ 0.05; **—*p* ≤ 0.01; ***—*p* ≤ 0.001; ****—*p* ≤ 0.0001.

**Figure 6 ijms-22-11970-f006:**
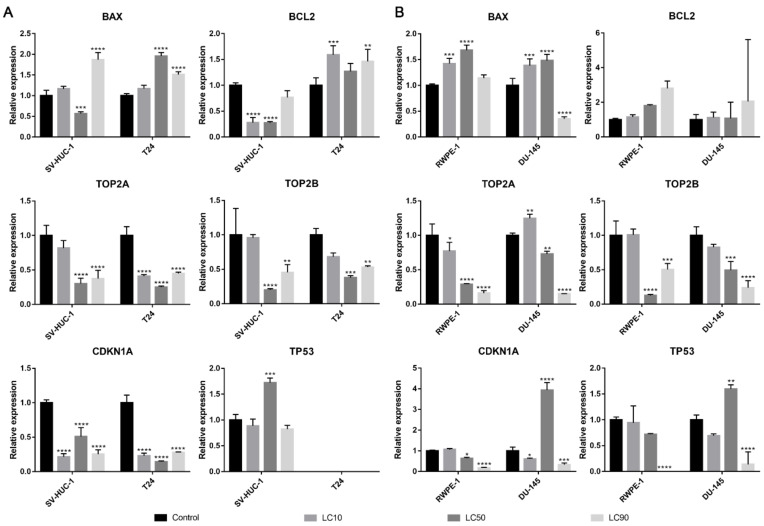
Molecular analysis of gene expression in bladder and prostate cell lines following incubation with ciprofloxacin. Experiment was performed using LC values of ciprofloxacin obtained after 24 h incubation. Relative gene expression of control was calculated as 1. (**A**)—Bladder cell lines. (**B**)—Prostate cell lines. SV-HUC-1—non-malignant human urothelium; T24—human bladder cancer; DU-145—human prostate cancer; RWPE-1—non-malignant human prostate epithelium. Statistically significant results compared to control were presented: *—*p* ≤ 0.05; **—*p* ≤ 0.01; ***—*p* ≤ 0.001; ****—*p* ≤ 0.0001.

**Figure 7 ijms-22-11970-f007:**
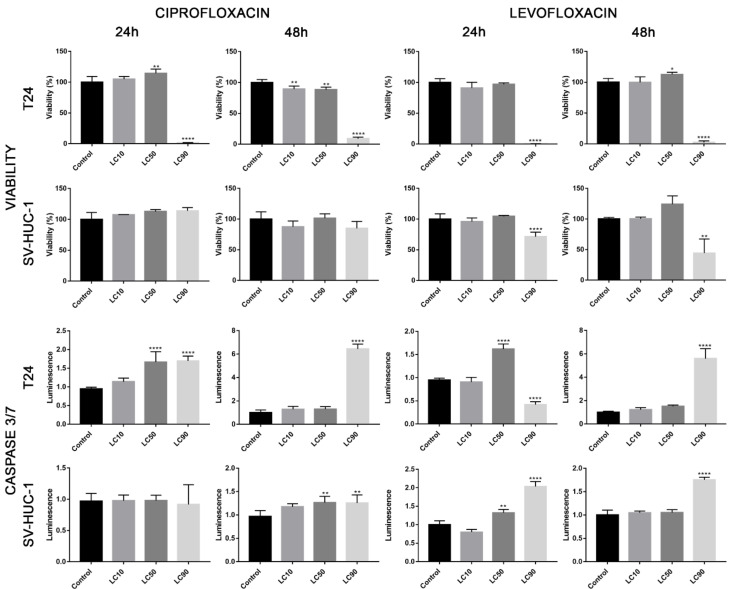
Three-dimensional (3D) cell culture assays following ciprofloxacin and levofloxacin treatment. Analysis of cell viability and activity of caspase 3/7 on spheroids generated from urothelial cell lines. Spheroids were treated by calculated LC concentrations obtained after 24 and 48 h incubation with ciprofloxacin and levofloxacin. Viability was presented as percentage compared to control, luminescence of caspases activity in control was calculated as 1. SV-HUC-1—non-malignant human urothelium; T24—human bladder cancer; DU-145—human prostate cancer; RWPE-1—non-malignant human prostate epithelium, LC—lethal concentration; statistically significant results compared to control were presented: *—*p* ≤ 0.05; **—*p* ≤ 0.01; ****—*p* ≤ 0.0001.

**Figure 8 ijms-22-11970-f008:**
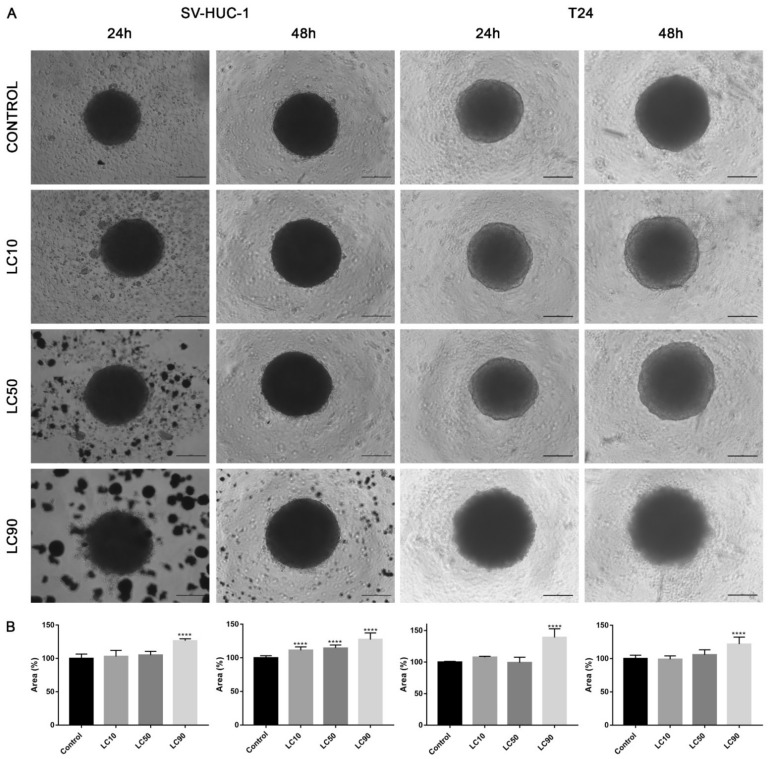
Morphology of urothelial cells in 3D cell culture following ciprofloxacin treatment. (**A**)—T24 and SV-HUC-1 cell lines treated with calculated lethal concentration obtained for ciprofloxacin, using an inverted light microscope (bar = 200 µm). (**B**)—Morphometric analysis of spheroids area. SV-HUC-1—non-malignant human urothelium; T24—human bladder cancer; LC—lethal concentration; statistically significant results compared to control were presented: ****—*p* ≤ 0.0001.

**Figure 9 ijms-22-11970-f009:**
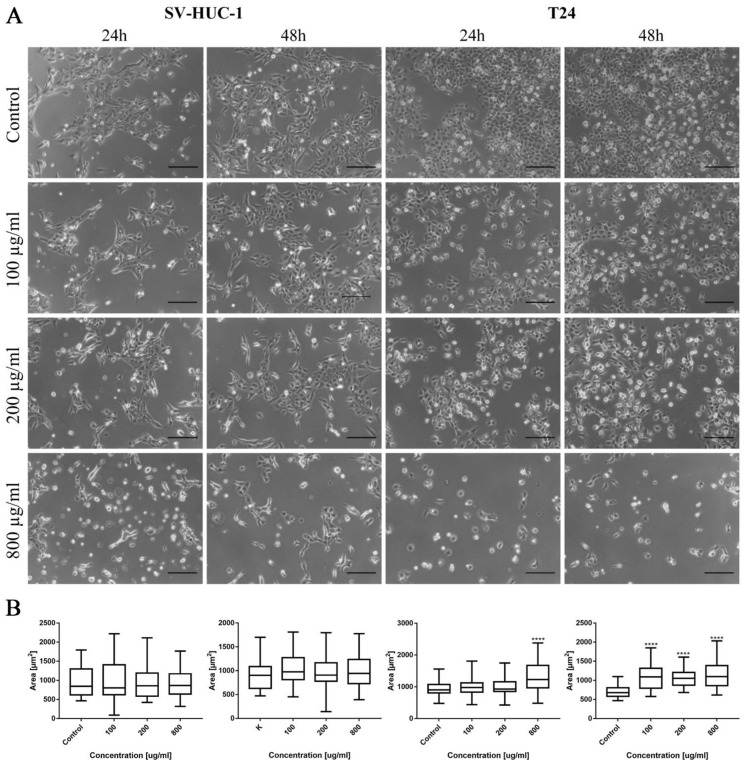
Bladder cell lines characteristic after levofloxacin treatment. (**A**)—Morphology of bladder cell lines after levofloxacin treatment. Cell shrinkage, rounding, and detachment were visible after levofloxacin treatment, especially in higher concentrations. Due to the lower toxic properties, morphological changes were less pronounced compared to ciprofloxacin. The addition of levofloxacin to cell culture did not cause crystals formation, using an inverted light microscope (bar = 200 µm). (**B**)—Morphometric analysis of cells area. SV-HUC-1—non-malignant human urothelium; T24—human bladder cancer; statistically significant results compared to control were presented: ****—*p* ≤ 0.0001.

**Figure 10 ijms-22-11970-f010:**
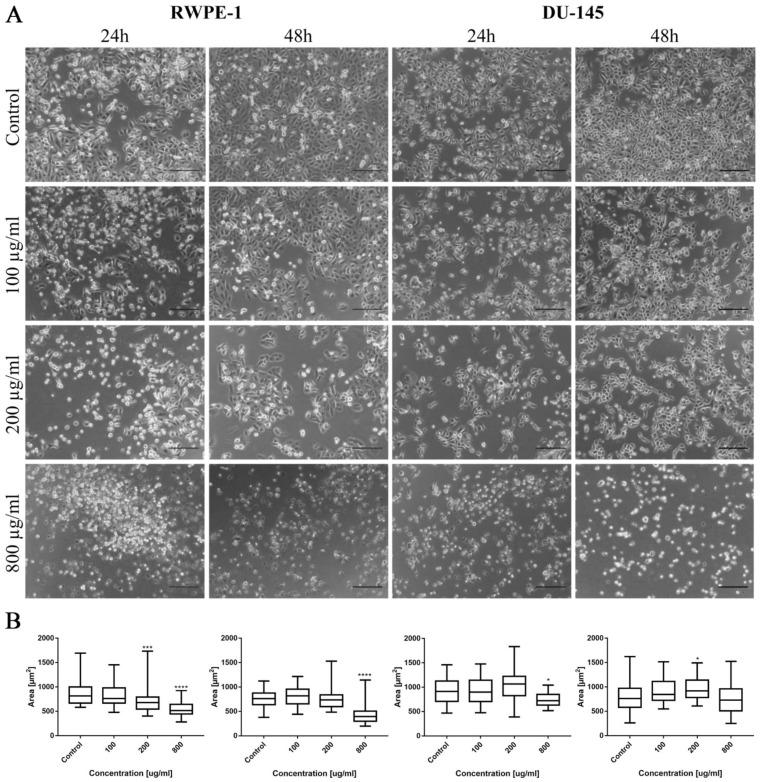
Prostate cell lines characteristic after levofloxacin treatment. (**A**)—Morphology of prostate cell lines after levofloxacin treatment. Cell shrinkage, rounding, and detachment were visible especially in higher levofloxacin concentrations. The addition of levofloxacin to cell culture did not cause crystals formation, which was observed using an inverted light microscope (bar = 200 µm). (**B**)—Morphometric analysis of cells area. DU-145—human prostate cancer; RWPE-1—non-malignant human prostate epithelium; statistically significant results compared to control were presented: *—*p* ≤ 0.05; ***—*p* ≤ 0.001; ****—*p* ≤ 0.0001.

**Figure 11 ijms-22-11970-f011:**
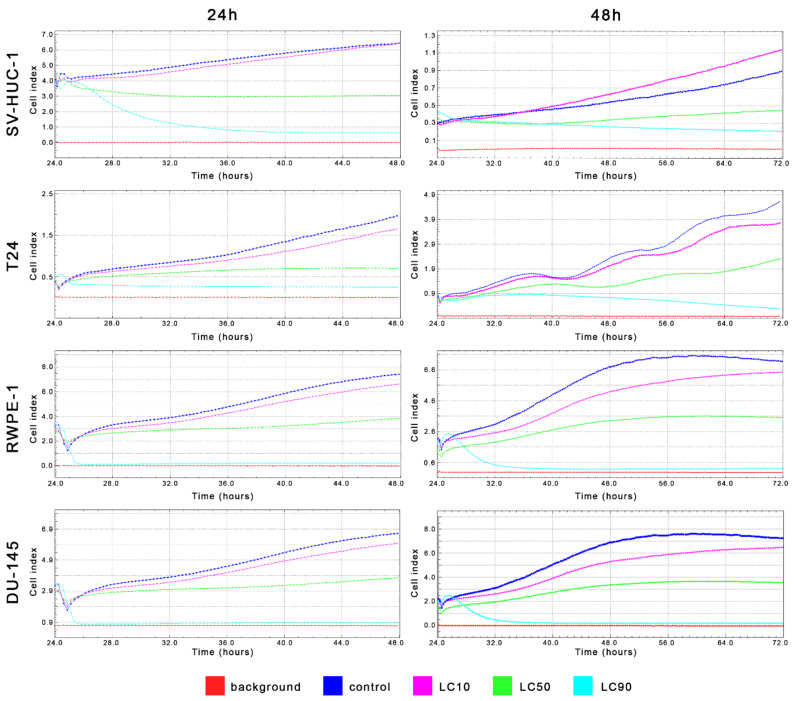
Real-time cell growth analysis of bladder and prostate cell lines after levofloxacin treatment. Data were measured every 30 min for 24 or 48 h. The obtained results confirmed that the LC values calculated using MTT assay actually cause a reduction of cells viability by 10, 50, and 90%. SV-HUC-1—non-malignant human urothelium; T24—human bladder cancer; DU-145—human prostate cancer; RWPE-1—non-malignant human prostate epithelium.

**Figure 12 ijms-22-11970-f012:**
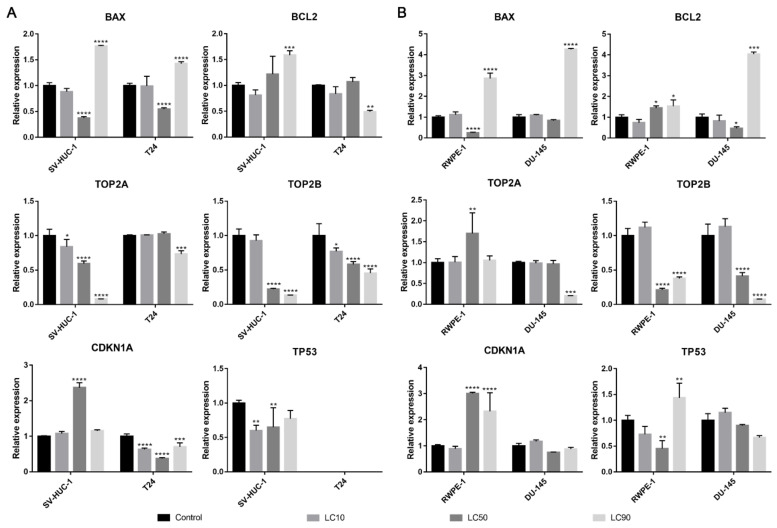
Molecular analysis of gene expression in bladder and prostate cell lines following incubation with levofloxacin. Experiments were performed using LC values of levofloxacin obtained after 24 h incubation. Relative gene expression of control was calculated as 1. (**A**)—bladder cell lines. (**B**)—prostate cell lines. Statistically significant results compared to control were presented: *—*p* ≤ 0.05; **—*p* ≤ 0.01; ***—*p* ≤ 0.001; ****—*p* ≤ 0.0001.

**Figure 13 ijms-22-11970-f013:**
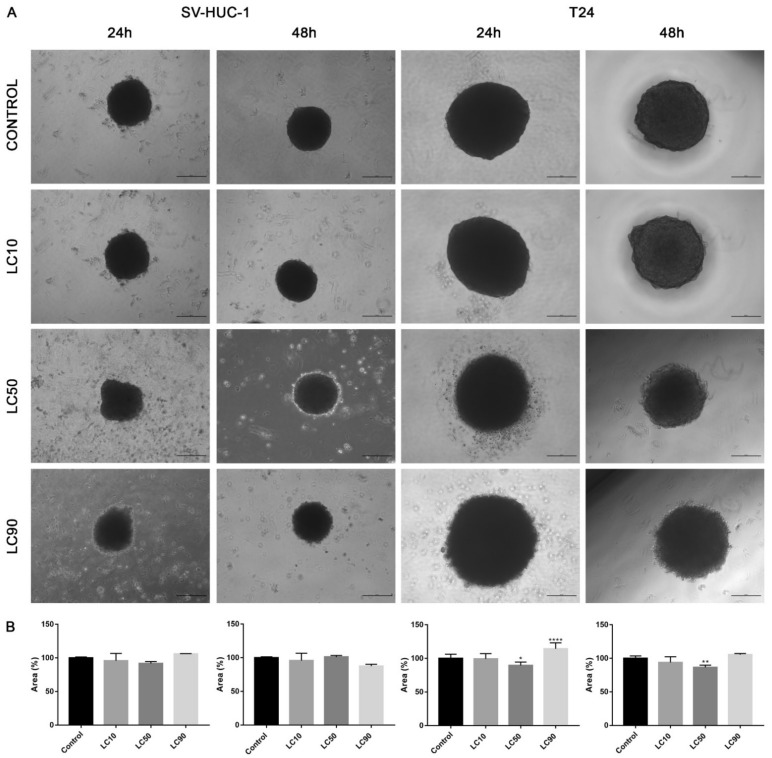
Morphology of bladder cells in 3D cell culture following levofloxacin treatment. (**A**)—SV-HUC-1 and T24 cell lines treated with calculated lethal concentration obtained for levofloxacin, using an inverted light microscope (bar = 200 µm). (**B**)—Morphometric analysis of spheroids area. SV-HUC-1—non-malignant human urothelium; T24—human bladder cancer; statistically significant results compared to control were presented: LC—lethal concentration; *—*p* ≤ 0.05; **—*p* ≤ 0.01; ****—*p* ≤ 0.0001.

**Table 1 ijms-22-11970-t001:** Lethal concentrations (LC, µM values) calculated for ciprofloxacin and levofloxacin after 24 and 48 h incubation with SV-HUC-1 (non-malignant human urothelium), T24 (human bladder cancer), DU-145 (human prostate cancer), and RWPE-1 (non-malignant human prostate epithelium) cell lines. With an asterisk (*), we marked values that were not achieved in MTT assay and were calculated theatrically from the curve equation.

Ciprofloxacin					
	LC [µM]	SV-HUC1	T24	RWPE-1	DU-145
24 h	LC10	167.77	15.42	64.10	95.25
	LC50	798.38	262.08	469.48	498.81
	LC90	3799.62 *	4559.71 *	4586.75 *	2611.92 *
48 h	LC10	113.42	60.96	75.51	47.08
	LC50	518.70	120.54	139.52	245.00
	LC90	2372.35	798.86	648.87	1290.64
**Levofloxacin**					
24 h	LC10	560.81	104.02	226.14	86.45
	LC50	2405.581	1146.59	1366.14	924.96
	LC90	4250.52 *	12,923.11 *	8252.81 *	6099.05 *
48 h	LC10	171.57	78.92	103.63	42.42
	LC50	1401.45	397.68	492.21	420.46
	LC90	2631.36 *	2021.85	2338.45 *	4166.97 *

## Data Availability

All data analyzed during this study are included in this published article.

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
