# Peer review of "Ciprofloxacin and Levofloxacin as Potential Drugs in Genitourinary Cancer Treatment—The Effect of Dose–Response on 2D and 3D Cell Cultures"

_ijms, 2021, doi:10.3390/ijms222111970_

Round 1

Reviewer 1 Report

The paper by Kloskowski et al describe the repositioning of Ciprofloxacin and levofloxacin for the treatment of bladder and prostate cancer. The authors show that these drugs have cytotoxic effect on bladder and prostate cancer cell lines compared to non malignant cells.

While these kind of studies are much needed to find new therapies, specially for aggressive and hard to treat cancers, such as the one studied in this manuscript, there are several points that needs to be addressed both on the technical side and conception of the study.

Major comments:

The paper needs english editing, as there are several grammar and spelling errors.

Throughout the text the authors use the term “normal” for non tumoral cell lines. Would it not be more correct to use non tumoral instead of “normal”?

What is the exact molecular mechanism and the target of these drugs in bladder or prostate cancer? The authors point out that the mechanism involved could be different from topoisomerase inhibition; could they speculate a bit more? They also discuss findings on Bcl2 proteins. Could they check for Bcl2 proteins expression levels in prostate cancer treated with these drugs vs non malignant cells?

Often in the field, drug synergism is explored as a mean to decrease the amount of drug used to obtain a higher cytotoxic effect. I think is important to check if there are is any synergism between the two drugs.

All the data presented are performed in 2d cultures. It would be interesting to see how organoids or spheroids react to the drugs used in this study.

The authors use MTT to measure viability. This is not entirely correct as MTT measure mitochondrial dysfunction. Are the authors sure that these drugs do not alter mitochondrial metabolism, and the effect they observe are related to that? It would be probably better to use other methods to check for viability.

Figure 1 and 2: The authors present light microscopy images, but they do not describe how these images were taken or processed in the materials and methods. Please add the respective section.

Moreover, even though differences can be observed by eye, a quantification is necessary to truly understand the effect of these drugs on cell morphology. I suggest the authors to stain cells with whole cell fluorescent dye (such as cell tracker green) to obtain numerical descriptors of morphology (Cell Profiler could be an option to obtain those). It is hard to describe and compare by eye those changes, since WT and cancer cells have different morphology from the start. I can also see that in control condition, T24 cell are already quite round and there several detached cells. Indeed, the authors claim there is a difference between control and 100ug/ml in t24 cells; I can’t really see a difference. At 800 ug/ul after 48 hours there are differences but with no quantification or statistical significance the claim of the authors are not fully valid. Same for figure 2.

Additionally, the concentrations used in this figure seems too high for me. The presence of crystals is a good indicator of that. As the authors point in the study, crystal formation is not feasible for a patient treatment.

Figure 3: Dose-response data need to be represented as curves and not as bar graphs. Is hard to see any sigmoidal behaviour with these graphs. I also don’t really understand the purposes of figure 3C, please explain.

Figure 4: no explanation of this figure is provided apart from “Real-time analysis of cell growth with use of calculated LC values confirmed results obtained with MTT assay”. Please explain this figure in details. What are the findings and what they mean?

Cell cycle table: This is hard to read, please provide bargraphs instead of a table.

Apoptosis results: please explain in materials and methods how live, early or late apoptotic and necrotic cells percentages are calculated (e.g. early apoptotic are AnnexinV positive, PI negative population). Can the authors also provide western blots or imaging quantification of caspase activation?

Figure 5: How the authors can establish differences between this figure and figure 1 or 2 with no quantification or statistical assessment?

Paragraph 3.2.2: this section is very confusing, please improve.

Figure 7: Again, no explanation is provided for growth assay.

Minor remarks:

Figure legends need a bit more text to fully explain the figure.

lines 28-29: sentence is not clear, please rephrase.

Line 31: please remove comma after accumulates

Line 39: sentence not clear, please rephrase

Line 44: please correct “malignances” to malignancies.

Line 46-49: Please correct English grammar of these sentences.

Line 53-53: which difficult procedures are the authors referring to?

Line 242-243: please rephrase this sentence as it is not clear.

Line 305: concentration is misspelled.

Line 360: The authors name a Bcl/Bax ratio. Which Bcl are they referring to?

Author Response

Answers to the Reviewers comments

Thank You for valuable comments and review of our manuscript. All comments prepared by Reviewers were included. All correction in manuscript were marked in red.

Reviewer 1:

The paper by Kloskowski et al describe the repositioning of Ciprofloxacin and levofloxacin for the treatment of bladder and prostate cancer. The authors show that these drugs have cytotoxic effect on bladder and prostate cancer cell lines compared to non malignant cells.

While these kind of studies are much needed to find new therapies, specially for aggressive and hard to treat cancers, such as the one studied in this manuscript, there are several points that needs to be addressed both on the technical side and conception of the study.

Major comments:

The paper needs english editing, as there are several grammar and spelling errors.

Manuscript was once more checked for grammar and spelling errors.

Throughout the text the authors use the term “normal” for non tumoral cell lines. Would it not be more correct to use non tumoral instead of “normal”?

According to Reviewer suggestion the term “normal was changed for “non-tumoral”.

What is the exact molecular mechanism and the target of these drugs in bladder or prostate cancer? The authors point out that the mechanism involved could be different from topoisomerase inhibition; could they speculate a bit more? They also discuss findings on Bcl2 proteins. Could they check for Bcl2 proteins expression levels in prostate cancer treated with these drugs vs non malignant cells?

According to Reviewer suggestion BCL2 gene expression was examined together with 5 other genes, including BAX, TOP2A, TOP2B, CDKN1A and TP53, on all tested cell lines. Probable mechanism of tested fluoroquinolones was discussed more extensive.

Often in the field, drug synergism is explored as a mean to decrease the amount of drug used to obtain a higher cytotoxic effect. I think is important to check if there are is any synergism between the two drugs.

Thank You for this comment. The aim of our study was to evaluate which of the most often prescribed fluroquinolones (levofloxacin or ciprofloxacin) has more promising properties for genitourinary cancer treatment. Additionally synergism of drugs is tested mainly when they differ in mechanism of action while in the case of both drugs tested in this study, the target is the same – topoisomerase.

All the data presented are performed in 2d cultures. It would be interesting to see how organoids or spheroids react to the drugs used in this study.

Additional experiments were performed on bladder cell lines. We decided to perform 3D cell culture analysis on bladder cell lined due to results obtained in 2D cell culture, which indicated greater potential of fluoroquinolones for bladder cancer treatment. 

The authors use MTT to measure viability. This is not entirely correct as MTT measure mitochondrial dysfunction. Are the authors sure that these drugs do not alter mitochondrial metabolism, and the effect they observe are related to that? It would be probably better to use other methods to check for viability.

MTT is recommended test for analysis of drug cytotoxicity, additionally we performed such analysis in our previous experiment. Molecular analysis performed in this study showed changes in BAX and BCL2 gene expression after treatment with both drugs which indicates influence of tested drugs on mitochondrial activity. However, MTT assay should be confirmed by other methods that is why we performed real-time cell analysis with calculated by MTT assay LC concentration, and this analysis confirmed that results of MTT assay are reliable.

Figure 1 and 2: The authors present light microscopy images, but they do not describe how these images were taken or processed in the materials and methods. Please add the respective section.

Moreover, even though differences can be observed by eye, a quantification is necessary to truly understand the effect of these drugs on cell morphology. I suggest the authors to stain cells with whole cell fluorescent dye (such as cell tracker green) to obtain numerical descriptors of morphology (Cell Profiler could be an option to obtain those). It is hard to describe and compare by eye those changes, since WT and cancer cells have different morphology from the start. I can also see that in control condition, T24 cell are already quite round and there several detached cells. Indeed, the authors claim there is a difference between control and 100ug/ml in t24 cells; I can’t really see a difference. At 800 ug/ul after 48 hours there are differences but with no quantification or statistical significance the claim of the authors are not fully valid. Same for figure 2.

According to Reviewer suggestion an additional description was added to material and methods section. For statistical analysis cell were measured using EPview 1.3 software, additional graphs were added to figures with cell images.

Additionally, the concentrations used in this figure seems too high for me. The presence of crystals is a good indicator of that. As the authors point in the study, crystal formation is not feasible for a patient treatment.

Thank You for this comment. We used such range of tested concentration in order to more reliable calculate three concentrations (LC0, LC50 and LC90) used in further analysis. Such range of drugs concentration was established on the basis of previous studies.

Figure 3: Dose-response data need to be represented as curves and not as bar graphs. Is hard to see any sigmoidal behaviour with these graphs. I also don’t really understand the purposes of figure 3C, please explain.

According to Reviewer suggestion dose-response graphs were presented as curves.

Figure 3C was presented in order to check correlation between cytotoxicity profiles of both tested drugs. Results of cells viability (in percentage) was compared together using Pearson comparison. On the basis of obtained results we can show presence or lack of similarities between ciprofloxacin and levofloxacin action on all tested cell lines and the differences in effect of each drug between cancer and non-tumoral cell line. The closer the value is to 1, the greater correlation (similarity) is present. Values less than 1 may indicate an advantage of one drug over another. 

Figure 4: no explanation of this figure is provided apart from “Real-time analysis of cell growth with use of calculated LC values confirmed results obtained with MTT assay”. Please explain this figure in details. What are the findings and what they mean?

Results presented on this figure confirmed that LC values calculated on the basis of results obtained in MTT assay actually reduce cells viability by 10, 50 and 90%.

Cell cycle table: This is hard to read, please provide bargraphs instead of a table.

The suggested correction has been made.

Apoptosis results: please explain in materials and methods how live, early or late apoptotic and necrotic cells percentages are calculated (e.g. early apoptotic are AnnexinV positive, PI negative population). Can the authors also provide western blots or imaging quantification of caspase activation?

Additional explanation was provided. Analysis of caspase3/7 and caspase 9 activities were performed.

Figure 5: How the authors can establish differences between this figure and figure 1 or 2 with no quantification or statistical assessment?

For statistical analysis cell were measured using EPview 1.3 software, additional graphs were added to figures with cell images.

Paragraph 3.2.2: this section is very confusing, please improve.

Section has been improved.

Figure 7: Again, no explanation is provided for growth assay.

Similar like in Figure 4, results presented on this figure confirmed that LC values calculated on the basis of results obtained in MTT assay actually reduce cells viability by 10, 50 and 90%.

Minor remarks:

Figure legends need a bit more text to fully explain the figure.

Figure legends have been improved.  

lines 28-29: sentence is not clear, please rephrase.

This has been corrected.

Line 31: please remove comma after accumulates

This has been corrected.

 Line 39: sentence not clear, please rephrase

This has been corrected.

Line 44: please correct “malignances” to malignancies.

This has been corrected.

Line 46-49: Please correct English grammar of these sentences.

This has been corrected.

 Line 53-53: which difficult procedures are the authors referring to?

 Meaning difficult procedures we meant long-drawn and costly processes of introducing new dug on the market. This sentence was modified.

Line 242-243: please rephrase this sentence as it is not clear.

This has been corrected.

Line 305: concentration is misspelled.

This has been corrected.

Line 360: The authors name a Bcl/Bax ratio. Which Bcl are they referring to?

This has been corrected.

Reviewer 2 Report

Kloskowski et. al., in their current study, try to validate the importance of repurposing the quinolones-based drug application on bladder and prostate cancer. However, designing and experimental approach of the study have some concerns.

  1. What is the basis of the concentration selection? The study is a bit more incremental than explorative compared to the previous studies.
  2. There is no basis for the application of drugs. Specially, like in prostate cancer, whether application of these drugs will be better at castration resistance phase or at the hormonal sensitive stage was not discussed. Whether it would be combined with ADT or it will work in AR independent fashion has not been studied.  How these drugs are incorporated within the cells needs to be clarified on the basis of disease progression.

Overall, the study is still naïve and needs more experimental evidence to clarify its application on GU cancer.     

Author Response

Answers to the Reviewers comments

Thank You for valuable comments and review of our manuscript. All comments prepared by Reviewers were included. All correction in manuscript were marked in red.

Reviewer 2:

Kloskowski et. al., in their current study, try to validate the importance of repurposing the quinolones-based drug application on bladder and prostate cancer. However, designing and experimental approach of the study have some concerns.

What is the basis of the concentration selection? The study is a bit more incremental than explorative compared to the previous studies.

We used such range of tested concentration in order to calculate three concentrations (LC0, LC50 and LC90) used in further analysis. Such approach allowed us obtain more reliable results. Such range of drugs concentration was established on the basis of previous studies, in which higher concentrations range were used. Our results showed that range between 25 and 800µg/ml is enough for LC10, LC50 and LC90 calculation.

Like we mention in manuscript, this is the first study directly comparing ciprofloxacin and levofloxacin. Additionally levofloxacin was for the first time tested against prostate cell lines. In the revised version of the manuscript w performed additional analysis including: molecular analysis was performed for the first time after levofloxacin action on bladder and prostate cells, both drugs were analyzed for the first time on 3D culture models.

There is no basis for the application of drugs.

In our opinion there are several points for consideration of fluoroquinolones as a potential drugs for genitourinary cancers treatment. Bot ciprofloxacin and levofloxacin are standard drugs used for treatment of genitourinary tract infection, fluoroquinolones accumulates in higher concentrations in urine and in prostate gland than in serum after oral and intravenous administration; in the case of bladder cancer calculated concentration are achievable to obtain in urine; in the literature we can find many evidences about anti-cancer properties of fluoroquinolones.

Specially, like in prostate cancer, whether application of these drugs will be better at castration resistance phase or at the hormonal sensitive stage was not discussed. Whether it would be combined with ADT or it will work in AR independent fashion has not been studied. How these drugs are incorporated within the cells needs to be clarified on the basis of disease progression.

The aim of our study was to check effect of ciprofloxacin and levofloxacin directly on prostate cells but not metastatic lesions. The effect of tested drugs on different stages of disease progression (including metastasis stage) was not our goal. We tested potential of both drugs in bladder and prostate cancer treatment. Our results showed, that in the case on prostate cells, calculated concentration (LC10, LC50 and LC90), which were effective in in vitro study are not achievable in prostate gland after oral and intravenous administration. That is why, compared to bladder cancer, application of levofloxacin and ciprofloxacin in prostate cancer has limited use. 

Overall, the study is still naïve and needs more experimental evidence to clarify its application on GU cancer.     

Like we mentioned above, we have performed additional analysis, which confirmed our hypothesis, that ciprofloxacin and levofloxacin can be considered as a potential drugs in supportive therapy of bladder cancer treatment.

Round 2

Reviewer 1 Report

The authors have improved the paper based on the suggestions made.

Reviewer 2 Report

I am accepting this modified form of the research article.